# Investigating the Variation in Leaf Traits Within the *Allium prattii* C.H. Wright Population and Its Environmental Adaptations

**DOI:** 10.3390/plants14040541

**Published:** 2025-02-10

**Authors:** Shuai-Shuai Zhang, Zi-Jun Tang, Kun Chen, Xiao-Jing Ma, Song-Dong Zhou, Xing-Jin He, Deng-Feng Xie

**Affiliations:** 1Key Laboratory of Bio-Resources and Eco-Environment of Ministry of Education, College of Life Sciences, Sichuan University, Chengdu 610065, China; 2State Key Laboratory for Quality Ensurance and Sustainable Use of Dao-di Herbs, National Resource Center for Chinese Materia Medica, China Academy of Chinese Medical Sciences, Beijing 100700, China

**Keywords:** *Allium prattii*, morphological characteristics, climatic factors, mountains of southwest China

## Abstract

Morphological and micro-morphological traits of characteristics serve as the cornerstone for species identification and taxonomy, and they also ensure the adaptive responses of species to specific environmental conditions. *Allium prattii* C.H. Wright is mainly distributed in the mountains of southwestern China (MSC) and adjacent regions, and exhibits pronounced variations in leaf morphology and micro-morphology across different growth environments, making it an ideal taxa to study species adaptation to diverse conditions. In this study, we conducted extensive field surveys, sample collections, and morphological experiments, amassing data on leaf morphological and micro-morphological traits from 45 populations of *A*. *prattii*. Specifically, we explored the differences in leaf morphology among populations and the patterns of geographical distribution. Consequently, we examined the correlation between seven climatic factors, longitude, latitude, and leaf morphological traits, and simulated the changes in the *A*. *prattii* distribution area during different historical periods. Our results indicate that all populations of *A*. *prattii* can be categorized into four distinct lineages, characterized by significant leaf morphological divergence and distinct geographical distribution patterns. Populations located in the Hengduan Mountains and neighboring regions demonstrated elevated coefficients of variation (CV) in leaf morphology. The correlation analysis between morphological traits and climatic factors highlighted substantial links between the density of stomata on the upper epidermis and environmental variables, as well as significant correlations between leaf length/width and geographical distribution (latitude and longitude). Simulations of the distribution area revealed that the distribution ranges of *A. prattii* underwent a significant fluctuation from the Last Interglacial Period (LIG) to the Last Glacial Maximum (LGM), the Mid-Holocene (MH), and the current period, accompanied by expansion of its potential distribution area in the future. These results underscore that the leaf morphology of *A*. *prattii* has significantly varied in response to climatic environmental factors across different regions, with a decrease in leaf width and an increase in stomatal density on the upper epidermis. The heterogeneous environment of the southwestern mountain region, characterized by variations in altitude, temperature, and precipitation, is the primary driver of morphological variation and geographical distribution patterns in *A. prattii* leaves. Our findings hold substantial scientific significance, shedding light on the evolutionary adaptation of species in the MSC and adjacent areas.

## 1. Introduction

Environmental heterogeneity is a pivotal factor influencing species evolution and ecological adaptation [1]. Within diverse habitats, a single species may encounter various environmental challenges, including low temperatures, drought, and low atmospheric pressure, each exerting a distinct influence on genetic diversity, population dynamics, and phenotypic traits [2,3,4]. Species counter environmental fluctuations by modifying their phenotypic traits or by adjusting gene expression and physiological metabolism, thereby maximizing their adaptability under specific environmental conditions [5]. Alpine environments, characterized by high altitude, significant temperature variations, and intense ultraviolet radiation, exhibit distinctive geographical and climatic complexity, making them quintessential examples of heterogeneous environments. In these settings, factors such as water availability, atmospheric pressure, and light intensity vary with altitude, prompting plants to display robust adaptive traits in both phenotype and genetic makeup [6]. The mountains of southwestern China (MSC) and adjacent regions, characterized by their intricate geography, stand out as a biodiversity hotspot and a climate change-sensitive area [7]. Investigating the phenotypic variations and alpine adaptability of species in these regions is of great significance for understanding the diversification and adaptive evolution of species in alpine environments.

Elevated altitudes curtail the growing season for plants and limit the abundance and activity of pollinating insects, which in turn reduces the resources available for growth and reproduction [8]. Studies have indicated that among 31 plant species on the Qinghai–Tibet Plateau, there is a notable decrease in the individual size and biomass of both nutritional and reproductive organs as altitude increases [9]. Furthermore, high-altitude species often display pronounced morphological differences compared to their low-altitude counterparts [10]. For example, research on *Taxus chinensis* (Pilger) Rehder and *Epilobium amurense* Hausskn. has highlighted significant variations in leaf morphology across different altitudes [11,12,13,14,15,16,17,18]. While in-depth research on phenotypic changes and environmental adaptability in alpine species continues to progress, the majority of studies have focused on non-sessile organisms, such as high-altitude humans, mammals, birds, and reptiles [19,20,21,22]. In contrast, research on the adaptability of sessile plants is scarce, with known cases such as *Crucihimalaya himalaica* (Edgew.) Al-Shehbaz, O’Kane & R. A. Price, *Alnus incana* (L.) Moench, *Fagus syvatica* L., and *Circaeaster agrestis* Maxim [9,11,13,14]. We posit that alpine plants, which cover vast altitude gradients and geographical ranges, are likely to show distinct morphological responses to their growth environments. Thus, a comprehensive morphological survey across the entire geographical distribution, coupled with an analysis of ecological factors, is crucial to uncover the morphological variations and adaptability of high mountain plants to external environmental pressures.

*Allium prattii* C.H. Wright (*Allium* L., Alliaceae) is a perennial herb distinguished by its unique, reticulated fibrous bulb scales and its possession of two (occasionally one or three) linear or oval leaves and is classified into subgenus *Anguinum* (G.Don ex Koch) N.Friesen [23]. This species enjoys a broad distribution, from the Himalayas across the southeastern edge of the Qinghai–Tibet Plateau and the Hengduan Mountains extending to the eastern slopes of the Qinling Mountains in China. Like other economically significant *Allium* crops such as garlic, onions, and leeks, *A. prattii* is not only rich in nutritional value but also a popular wild vegetable in highland regions. The entire plant is a treasure trove of sterols, sulfides, and flavonoids, which are known for their potent antibacterial, antioxidant, and therapeutic properties against a variety of diseases [24,25]. Recognized for its dual role as a medicinal and edible plant, *A. prattii* represents a valuable resource with substantial development and utilization potential.

Through preliminary field surveys, we discovered that this species is distributed at a minimum altitude of approximately 1600 m, extending up to nearly 5000 m, offering a substantial altitudinal span that is pivotal for examining the species’ environment adaptation and evolution at high elevations. Furthermore, our field observations have highlighted significant morphological diversity among individuals of *A. prattii* populations across a range of altitudes. With increasing altitude, the habitats changed and the plants tended to be more dwarfish, and their leaf shape shifted from elliptical or oval at lower elevations to linear or narrow at higher elevations, with leaf thickness also augmenting at higher altitudes (Figure 1). Many studies have indicated that species can evolve specialized physiological and morphological adaptations in response to heterogeneous environments, enhancing their capacity to adapt to external conditions [26,27,28,29]. Therefore, we hypothesize that the significant variation in leaf morphology among *A. prattii* populations may be an important reflection of their adaptation to environmental heterogeneity at different altitude gradients. Previous studies have predominantly concentrated on the subgenus *Anguinum*, focusing on interspecific morphology, cytology, and molecular systematics [30,31,32,33,34,35]. These studies have shed light on the phylogenetic positioning and intricate species relationships within the subgenus *Anguinum*. However, research on intraspecific leaf morphological variation among *A. prattii* populations and the relationship between this variation and their environmental adaptation has not yet been conducted, underscoring an urgent need for such investigative efforts.

In this study, we engaged in comprehensive field surveys, collected samples, and conducted morphological studies to gather leaf morphological data from 45 populations of *A. prattii*. We statistically analyzed the variation in leaf morphological traits across populations and performed comparative and clustering analyses. Additionally, we investigated the correlations between populations in terms of leaf morphological variation and the climatic environment, as well as modeled the shifts in the species’ distributional ranges across different historical epochs. Our aims were to determine the following: (i) whether there is a discernible pattern of leaf morphological variation at the population level for *A. prattii*; (ii) whether climatic environmental factors play a role in the leaf morphological variation observed at the population level of this species; and (iii) whether the distribution range of the *A. prattii* species has experienced substantial shifts in response to geo-climatic fluctuations across various historical epochs.

## 2. Results

### 2.1. Leaf Morphological and Micro-Morphological Characteristics

We selected 5–8 representative individuals from each population for morphological measurements and leaf micro-morphological experiments, accumulating a total of 268 plant specimens. The morphological measurement results of the leaves were statistically analyzed, which showed that only 11 populations had stomata on the upper epidermis of the leaves, including A08, A09, A10, A11, and A24, etc. (Table 1). Stomata were ubiquitous on the lower epidermis across all populations; however, there was considerable variation in stomatal density. Notably, population A34 exhibited the highest density with an average of 90.63 stomata, whereas population A05 had the lowest with an average of 9.38 stomata. Furthermore, variations in leaf length and width were observed among different populations. Specifically, populations A08 and A09 had leaves with an average length exceeding 35 cm and an average width of over 2 cm. In contrast, populations A04 and A41 had leaves with an average length of less than 15 cm and an average width under 1.7 cm. Given the variation within the population, which was measured using the coefficient of variation, we found that the coefficient of variation for leaf length in 17 populations was below 10%. However, in three populations (A03, A31, and A42), this coefficient exceeded 30%. Notably, population A31 exhibited an exceptionally high coefficient of variation, reaching 47.71% (Table 1). Regarding leaf width, only population A28 exhibited a coefficient of variation below 10%. In contrast, 14 populations displayed a coefficient of variation for leaf width exceeding 30%, with population A37 standing out for its remarkably high coefficient of variation at 49.42%. These results indicate that populations situated at the margins of the distribution range display more pronounced leaf morphological variation, such as populations A31, A37, and A03, and populations distributed in MSC and adjacent regions exhibit high CV values.

### 2.2. Cluster Analysis and ANOVA Test for Morphological Data

Utilizing UPGMA hierarchical clustering analysis, we categorized the 45 populations of *A. prattii* into four distinct lineages. By applying inter-group linkage methods to all morphological data for the hierarchical clustering of these populations, a dendrogram was derived. We then visualized the geographic distribution of the *A. prattii* populations according to the clustering outcomes (Figure 2). Our analysis revealed that one lineage is predominantly situated in the Himalayas, and has been designated as the west lineage. The second lineage encompasses areas in Shaanxi, Henan, and Hubei, earning it the name east lineage. The third lineage is concentrated in the southern part of Qinghai and the northwestern reaches of Sichuan, hence it is termed the north lineage. Lastly, the fourth lineage, spread across western Sichuan, northwestern Yunnan, and eastern Xizang, is geographically designated as the central lineage. Based on the ANOVA test, we discovered that the west lineage exhibited significantly larger leaf length and width indices compared to other lineages, while their stomatal density index was markedly lower. The north lineage demonstrated leaf length and width indices that were substantially greater than those of both the central and east lineages, and their stomatal density index was also notably higher than that in the central and east lineages (Appendix A).

### 2.3. Principal Component Analysis of Leaf Traits

The principal component analysis (PCA) of leaf traits for *A. prattii* reveals that the first and second principal components explain 39% and 32.7% of the variance, respectively (Figure 3A), leading to the selection of these two components for subsequent analysis. Notably, average leaf width (Average_width) and average leaf length (Average_length) significantly contribute to the first principal component (Figure 3B), whereas the density of stomata on the upper epidermis (SD_upper) and lower epidermis (SD_under) are major contributors to the second principal component (Figure 3C). When considering both principal components, the density of stomata on both epidermis layers and leaf width emerge as the most influential factors (Figure 3D).

The correlation analysis of four leaf traits against four principal components reveals that leaf length, width, and stomatal density on the upper and lower epidermis have diverse correlations with the components Dim1 through Dim4, as illustrated in Figure 3E–G. The PCA clustering outcomes for these leaf traits show that Dim1 and Dim2 explain 39% and 32.7% of the variance in *A. prattii*, respectively (Figure 3H), with each sample holding statistical significance. Notably, the first principal component (Dim1) is predominantly defined by leaf length and width, and the second principal component (Dim2) is chiefly linked to the stomatal density on both epidermis layers. A scatter plot utilizing Dim1 and Dim2, with samples clustered hierarchically, indicates that the north lineage possesses higher scores for both Dim1 and Dim2, implying larger leaves with a higher stomatal density. Conversely, the west lineage exhibits higher Dim1 and lower Dim2 values, suggesting that it features larger leaves with a lower stomatal density (Figure 3I).

### 2.4. Principal Component Analysis of Environmental Factors

We selected the standardized data of the seven main environmental factors for principal component analysis (Appendix A), extracting components based on the criteria of a cumulative value greater than 75% and an eigenvalue greater than 1. As depicted in the scree plot (Figure 4A), the first two principal components account for 52.8% and 23.9% of the variance, respectively, amassing a cumulative contribution of 76.7%. This suggests that these two components encapsulate the principal information of the seven environmental factors, hence we chose to retain two principal components for subsequent analysis. Within the principal component analysis, a higher contribution rate signifies a more substantial impact of the factor on the respective component. The first principal component, with a variance contribution rate of 52.8%, plays a pivotal role in our analysis, significantly influenced by annual precipitation (Bio12), precipitation of the driest month (Bio14), isothermality (Bio3), and the mean temperature of the warmest quarter (Bio10) (Figure 4B). The second principal component, contributing 23.9% to the variance, is notably shaped by the temperature seasonality coefficient (Bio4) and the annual temperature range (Bio7) (Figure 4C). When considering both principal components, the temperature seasonality coefficient (Bio4), annual temperature range (Bio7), isothermality (Bio3), and annual precipitation (Bio12) emerge as the most influential factors (Figure 4D).

The correlations and squared correlations between the seven environmental variables and the four principal components are shown in Figure 4E,F. Alt, Bio3, Bio10, Bio12, and Bio14 are significantly correlated with Dim1, with Alt and bio3 showing negative correlations and Bio10, Bio12, and Bio14 showing positive ones. Dim2 is strongly and positively associated with Bio4 and Bio7. Altitude and Bio14 are strongly correlated with Dim3, while Alt and Bio10 have a strong positive correlation with Dim4. The contribution of variables to the principal components is depicted in Figure 4G: Bio3, Bio12, Bio14, and Bio10 contribute most to Dim1; Bio4 and Bio7 contribute most to Dim2; Alt contributes most to Dim3; and Alt and Bio10 contribute most to Dim4.

The principal component analysis (PCA) of the seven selected environmental factors discloses that the first and second principal components (PC1 and PC2) explain 52.8% and 23.9% of the variance, respectively (Figure 4H). The PCA of the normalized environmental data reveals that the first principal component (Dim1) is largely influenced by Bio12, Bio3, Bio14, and Bio10, whereas the second principal component (Dim2) is primarily defined by Bio4 and Bio7. A scatter plot utilizing Dim1 and Dim2, with samples grouped based on the outcomes of hierarchical clustering, was visualized (Figure 4I). The results indicate that the east lineage is associated with higher Dim1 values, while the central lineage is associated with lower Dim1 values. The north lineage is characterized by lower Dim1 and higher Dim2 values, and the west lineage is marked by lower values for both Dim1 and Dim2. These findings suggest that the east lineage is defined by lower altitude and isothermality, along with greater precipitation and higher average temperatures during the warmest season. Conversely, the central lineage is characterized by higher altitude and isothermality, coupled with lower average temperatures in the warmest season. The north lineage is noted for its higher altitude and isothermality, reduced precipitation, and more extensive annual temperature range, pointing to considerable temperature variability. In contrast, the west lineage is characterized by higher altitude and isothermality, with less temperature variation.

### 2.5. Correlation Between Leaf Traits and Geographical Climatic Factors

To assess the relationship between variation in leaf traits and environmental factors, we conducted a Spearman correlation analysis between leaf traits and major environmental factors (Figure 5). The findings reveal that the density of stomata on the lower epidermis is not significantly correlated with any environmental factors. However, the density of stomata on the upper epidermis, the average leaf length, and the average leaf width display more distinct correlations with environmental factors. Notably, the density of stomata on the upper epidermis correlated positively with altitude (Alt) (*p* < 0.01), annual temperature range (Bio7) (*p* < 0.01), and temperature seasonality coefficient (Bio4) (*p* < 0.05), and negatively correlated with the mean temperature of the warmest quarter (Bio10) and annual precipitation (Bio12) (*p* < 0.01). Conversely, leaf width significantly positively correlated with the warmest quarter (Bio10) and the precipitation of the driest month (Bio14) (*p* < 0.01), and significantly negatively correlated with altitude (Alt) and latitude (*p* < 0.05). The leaf length significantly positively correlated with longitude (*p* < 0.05).

### 2.6. Ecological Niche Distribution Modeling

This research forecasts the potential distribution zones of *A. prattii* across various historical timeframes, achieving an average AUC value exceeding 0.95 for 50 test datasets and maintaining AUC values above 0.965 for training datasets (Appendix A). Within the spectrum of seven environmental variables, the temperature seasonality coefficient (Bio4) emerges as the primary factor influencing the distribution of *A. prattii*, contributing 44.9% to the model, with the mean temperature of the warmest quarter (Bio10) a close second at 30.8%. The remaining variables, annual precipitation (Bio12), annual temperature range (Bio7), altitude, isothermality (Bio3), and precipitation of the driest month (Bio14), account for 8%, 6.8%, 5.8%, 2.5%, and 1.3% of the distribution pattern, respectively (Appendix A).

Comparing the potential suitable distribution areas of *A. prattii* from the LIG to the LGM to the current period, the results primarily show a trend of contraction, with reductions mainly occurring in southern Sichuan, northern Yunnan, southern and eastern Qinghai, and southwestern Xizang, while most suitable habitats remained unchanged and expanded in eastern Xizang, Shaanxi, and Gansu. From the LIG to the LGM, there was an obvious contraction of suitable distributions, mainly in northwest Sichuan, southern Qinghai, and the Himalayas (Figure 6A). From the LGM to the Mid-Holocene (MH), there was a significant expansion of suitable distributions, extending outward around the HDM areas, such as northwest Sichuan and southern Xizang (Figure 6B). From the MH to the current period, there was a significant contraction in the Himalayas, and the contraction also slightly occurred in northeast Yunnan (Figure 6C). From the LIG to the LGM to the MH and then to the current period, it can be observed that the suitable distribution areas of *A. prattii* exhibit a significant fluctuation.

Forecasts for the shifts in the suitable distribution areas of *A. prattii* across the forthcoming periods (2021–2040 and 2041–2060) indicate that under the SSP245 scenario, the species’ habitat is expected to predominantly contract between 2021 and 2040, with slight reductions around the HDM regions, such as southern Qinghai (Figure 6D). From 2041 to 2060, under the same scenario, a slight distribution expansion of *A. prattii* is expected to occur, mainly in southeast Qinghai (Figure 6E). In essence, the suitable distribution area for *A. prattii* is projected to decrease from the present to 2021–2040, and it is expected to rebound and increase from 2041 to 2060. Consequently, it is likely that the potential suitable distribution area for *A. prattii* will expand in the future.

## 3. Discussion

### 3.1. Morphological Variation Among Different Lineages and Potential Drivers

The micro-morphological traits of the leaf epidermis serve as a foundational criterion for classifying *Allium* species [36]. These traits mirror the external environmental characteristics of plants and are intricately linked to the ecological factors of their habitats. In our study, we utilized a light microscope to observe and document the stomatal density within the leaf epidermis, revealing both uniformity and variation across diverse populations. Typically, the reduction or loss of stomata is an adaptation to arid conditions, representing a strategic response to water scarcity [37,38]. Our research discovered that in 45 populations of *A. prattii*, the upper epidermis was devoid of stomata in the majority of cases. This absence, in conjunction with existing studies [39], is indicative of the dry and cold habitat condition of *A. prattii*, which varies in altitude and encompasses a range of habitat types. The lack of stomata in the upper epidermis is fundamentally attributed to genetic determinants. However, in a minority of *A. prattii* populations, a modest number of stomata were present in the upper epidermis, albeit at a lower density than in the lower epidermis (Appendix A). Based on cluster analysis, we noted a distinct divergence in morphological traits among various *A. prattii* populations across different geographical areas, with the west lineage in the Himalaya region showing particularly pronounced differences (Figure 3). Moreover, despite the north and central lineages being in close proximity, they have clearly evolved into two distinct groups. This suggests that the north lineage has experienced changes in leaf characteristics and has morphologically diverged from the central populations under similar environmental conditions. Additionally, even though the central and east populations are geographically intermingled, there are evident variations in leaf morphological traits between them, which we hypothesize may stem from disparities in the micro-environments of these populations.

Leaves, as the pivotal site for material and energy transformation in plants, play a significant role in plant growth, development, and tolerance to shade [40]. They are the most reflective of a plant’s growth status and its resource utilization strategies. Hence, investigating variations in leaf traits is anticipated to uncover the adaptive strategies of plants to their environments. A multitude of studies has delved into the correlation between leaf traits and environmental factors, as well as the phenotypic plasticity across a spectrum of plants, including *Abies georgei* var. *smithii* [41], red sand [42], *Phoebe bournei* [43], and black sagebrush [44], shedding light on how plants modify their leaf functional traits to thrive in diverse environments. Research has established that the distribution patterns of stomata on leaves are predominantly governed by genetic factors [45]. In species possessing stomata on both leaf surfaces, the distribution on the lower epidermis tends to be more consistent than on the upper epidermis [46,47,48].

In our study, we noted that the upper epidermis of leaves from the north branch populations of *A. prattii* presented stomata, and the stomatal density on the lower epidermis was considerably higher than in other populations. The north lineage’s geographical region is characterized by high altitude, scant precipitation, and substantial temperature fluctuations (Figure 4 and Figure 5), prompting us to hypothesize that this phenomenon may stem from cold and drought stress. This aligns with the conclusions reached by Yu et al. [49] and Gao et al. [50]. The presence of stomata on the upper epidermis enhances *A. prattii*’s adaptation to cold and arid conditions, highlighting a crucial facet of its environmental adaptability. Furthermore, the interplay between leaf traits and the environment is shaped by a confluence of biotic and abiotic factors [51]. Leaf traits are influenced not only by meteorological elements, such as temperature, precipitation, and light [52], but also by geographical constraints, including topography, altitude, and slope [53], as well as nutritional status, disturbances like fire, and grazing intensity [54,55]. We thus posit that the emergence of stomata on the upper epidermis and the heightened stomatal density on the lower epidermis in the northern branch populations could be attributed to biotic or abiotic stressors, such as CO_2_ concentration and light, necessitating further experimental validation.

We also discovered that as latitude and altitude increase, the leaf width of *A. prattii* gradually decreases, and a significant positive correlation was detected between leaf length and longitude (Figure 5). Our field observations also clearly demonstrate that the leaf width of *A. prattii* narrows progressively with increasing altitude. For example, populations in alpine stone fields and shrub habitats exhibit significantly narrower leaves than those in forest understories. The geographical variation in plant leaf morphology is a complex phenomenon that is influenced by a multitude of factors such as longitude, latitude, altitude, and soil conditions. In particular, in some arid and semi-arid regions, leaf length tends to exhibit a significant positive correlation with longitude [56]. Other studies have shown that in higher latitudes with colder and drier climates, plant leaves tend to be shorter and narrower [57], and in western China, latitude can explain up to 75% of leaf morphological variation [58], which is consistent with our findings. Moreover, we observed that most of the *A. prattii* populations located in the Hengduan Mountains and neighboring regions consistently demonstrated elevated coefficients of variation (CV) in leaf morphology. Notably, those populations situated at the margins of the distribution range exhibited the most pronounced leaf CV values, suggesting a possible correlation with the distinctive climatic conditions of the Hengduan Mountains and its surrounding areas. *A. prattii* is predominantly found in the MSC and adjacent areas, marked by high altitudes, significant temperature variations, and intense ultraviolet radiation. Variables such as water heat, atmospheric pressure, and light fluctuate with altitude, leading to variations in population structure and phenotypic diversity among species [2,3,4]. This pattern has been corroborated across various taxa in the region, encompassing humans, mammals, birds, and certain plants [11,12,19,21,22]. Consequently, we hypothesize that the heterogeneous environment of the southwestern mountains, with its variations in altitude, temperature, and precipitation, is the principal driver of morphological variation and geographical distribution patterns in *A. prattii* leaves

### 3.2. Historical Distribution Fluctuations of A. prattii Populations

From historical periods to the present, we have observed a cyclical pattern in the distribution area for *A. prattii*, which initially contracted, then expanded, and subsequently contracted again (Figure 7). The distribution area reached its peak during the Mid-Holocene (MH), possibly due to the climatic conditions being most favorable for its growth during this period. In general, there has been an increasing trend in the suitable distribution area for *A. prattii*. The MSC and their adjacent regions, characterized by their high altitudes and distinctive geographical locations, have been subject to frequent and extensive ancient glaciations, resulting in a rich legacy of glacial relics [59]. These areas are sensitive and responsive to local or global climate changes and have become hot spots for studying glacial development and geomorphological evolution under global climate change scenarios [60,61,62,63]. We forecast the potential distribution patterns of *A. prattii* during three historical periods: the LIG, the LGM, and the MH. The LIG was one of the warmest interglacials of the past 800,000 years, with climatic features resembling the long-term climate change trends anticipated for the future [64,65]. The LGM was marked by cold, dry, and inhospitable conditions worldwide, with the Tibetan Plateau covered in ice and numerous small mountain glaciers emerging in Southeast Asia. The harsh and arid climate of this period resulted in a diminished plant distribution and a shift towards lower altitudes, aligning with our predictive outcomes. The MH period was warmer than the present day, with a warm and humid climate that fostered robust vegetation growth. This corresponds with the significant expansion trend of the suitable distribution area for *A. prattii* from the LGM to the MH period as modeled in this study.

Over the past few decades, the southeastern area of the MSC regions has transitioned to warmer and drier conditions, in contrast to the northwestern regions, which have grown warmer and more humid. These shifts have precipitated notable alterations in the vegetation across these zones [66,67]. As global warming persists, the potential for profound climatic transformations looms, with the southwestern mountainous regions, known for their acute sensitivity to climate fluctuations, anticipated to undergo even more marked changes [68,69]. Given that these areas, along with their environs, constitute the primary habitat for *A. prattii*, they are poised to bear the brunt of climate alterations, potentially reshaping the plant’s potential suitable distribution patterns. This study’s findings, leveraging climate data for two prospective periods under the SSP245 scenario (2021–2040 and 2041–2060) and employing the MaxEnt model for predictions, corroborate this assertion. Between 2021 and 2040, under the SSP245 scenario, the habitat conducive to *A. prattii* was found to primarily constrict in areas such as southern Qinghai, Tibet, northern and southern Sichuan, northern Yunnan, Shaanxi, Chongqing, Hebei, and other locales, registering a slight reduction from the present day. Nevertheless, the regions with optimal living conditions for the species saw a substantial 48.01% increase, concurrent with a rise in average altitude. From 2041 to 2060, under the same scenario, the suitable distribution area for *A. prattii* is projected to broaden, particularly in southeastern Qinghai, Tibet, Shaanxi, Sichuan, and adjacent regions, without any significant shift in average altitude. This suggests that forthcoming climatic conditions may be more favorable for the growth of *A. prattii*. Despite the current expansive distribution range of *A. prattii*, the escalation of human activities, the exacerbation of environmental concerns, and the accelerated rate of species extinction are posing unprecedented risks to global biodiversity [70,71]. In the midst of advancing scientific research, it is imperative that we intensify conservation measures to avert the devastation of wild populations of *A. prattii*.

## 4. Materials and Methods

### 4.1. Field Surveys and Sampling

From 2021 to 2024, we conducted field expeditions and collected samples, amassing a total of 45 populations of *A. prattii*. The sampling sites spanned across various regions, including Xizang, Sichuan, Qinghai, Yunnan, Shaanxi, Hubei, Henan, and more. The detailed sampling information is listed in Appendix A.

### 4.2. Morphological Data Measurement and Standardization

Specimens of *A. prattii* collected from the field were photographed, and the images were imported into MATO software [72] to measure the length, width, and area of the leaves, as shown in Figure 2. Consistent and healthy leaves from *A. prattii* plants were selected, and epidermal cell slides were prepared using sodium hypochlorite maceration. The leaves were treated with pure sodium hypochlorite for 5–10 min, after which the upper and lower epidermal layers were extracted. Leaf epidermal characteristics were observed under an optical microscope (Nikon, Yuansong, Chengdu, China), capturing photographs of the epidermis within the field of view under a 20x objective lens. Finally, the photographs were imported into MATO software [72] for stomatal count, determining the stomatal density on the upper and lower epidermis of the leaves. Data recording and preliminary statistics were conducted using Excel software. The leaf morphological variation within each population was measured using the coefficient of variation (CV), which was calculated as average values/standard deviation * 100%. Additionally, in R version 4.4.1 [73], the “scale” function was utilized to standardize the measured leaf trait data and obtained geographical information, and the standardized morphological data of the leaves were exported for subsequent analysis.

### 4.3. Cluster Analysis and Morphological Difference Test Among Groups

We conducted UPGMA hierarchical clustering analyses on leaf morphological traits using R version 4.4.1 [73]. Euclidean distances were calculated, and the sum of the squared deviations method (ward.D2) was employed to determine the inter-branch distances. The resulting dendrograms from the hierarchical clustering were plotted with adjusted color coding. To enhance the interpretability of the clustering outcomes, we utilized ArcGIS 10.8 software to create population distribution maps, which facilitated the visualization of the geographic distribution patterns across various populations. The R packages “FactoMineR”, “Hmisc”, and “factoextra” were employed in this analysis [74,75,76]. Moreover, based on the results of the cluster analysis, we conducted significance tests for differences in morphological characteristics among populations across groups. Considering the homogeneity of morphological variation, the ANOVA statistical method [77] was employed to detect the degree of morphological differences between groups. The R packages “dplyr”, “ggplot2”, and “multcompView” were used for the analysis.

### 4.4. Filtration of Climatic Factors

The environmental dataset included 19 bioclimatic variables (Appendix A), each interpolated at a 30 arc-minute resolution, derived from the WorldClim database (version 1.4, available at http://www.worldclim.org (accessed on 12 March 2024)). Additionally, an elevation layer was integrated from the SRTM database. For both current climate and elevation data, we calculated pairwise Pearson’s correlation coefficients (r) using ENMTOOLS version 1.4.3 [78] as described by Dormann et al. [79]. Variables with a correlation coefficient greater than 0.7 with two or more other variables were omitted to prevent multicollinearity. The distribution points of *A. prattii* were then entered into MAXENT. Subsequently, we employed a hierarchical partitioning method [80] via the R package HIER.PART [81] to ascertain which variable had the most significant independent effect on the data. Model efficacy was assessed by comparing the AUC (area under the receiver operating characteristic curve) values, which range from 0 to 1, with a score of 1 indicating perfect discrimination, as elucidated by Fielding and Bell [82]. Based on the outcomes of the Pearson’s correlation analysis and the contribution assessment for each climatic factor, we ultimately selected 7 environmental variables for subsequent analysis, which are elevation (Alt), isothermality (Bio3), temperature seasonality (Bio4), annual temperature range (Bio7), mean temperature of warmest quarter (Bio10), annual precipitation (Bio12), and precipitation of the driest month (Bio14).

### 4.5. Principal Component Analysis

We utilized the “FactoMineR” and “factoextra” packages in R version 4.4.1 [70], to perform a principal component analysis (PCA) using leaf morphological data and the above seven selected environmental factors. Principal components were extracted based on the criteria of a cumulative contribution greater than 75% and an eigenvalue greater than 1, followed by visualization plotting. Additionally, the “psych” and “reshape2” packages in R [83,84] were employed for correlation analysis between seven environmental factors, longitude, latitude, and *A. prattii* leaf morphological characteristics, and the “pheatmap” package was used to generate a heatmap [85].

### 4.6. Simulation of the Potential Distribution Area

We integrated environmental factor data spanning historical (the Last Interglacial Period, the Last Glacial Maximum, and the Mid-Holocene), current, and future periods (up to the year 2050), along with the coordinates of collection points, into Maxent 3.4.1 software [86] to simulate and forecast suitable distribution areas for each respective period. To bolster the precision of our models, we leveraged the “ENMeval” package in R version 4.4.1 [87] for the optimization of Maxent model parameters. We configured the model to run 100 replicated simulations, with each selecting 25% of the distribution points at random for validation and AUC determination. The Regularization multiplier was adjusted to 0.5, and we opted for the “Bootstrap” method for replicated runs. For our predictions, we chose the “Linear features” + “Quadratic features” setting, while all other parameters were left at their default values. The outcomes for suitable distribution areas were derived from the mean of 100 simulations.

Employing the Maximum Training Sensitivity and Specificity (MaxSS) approach, we established a threshold value of 0.323064. Following this, we transformed the model’s asc-formatted suitable distribution area results into GeoTiff format using ArcGIS (https://doc.arcgis.com/en/, accessed on 9 March 2024). We then utilized ArcGIS’s “Reclassify” tool to sort the simulation outcomes into categories: highly suitable areas (above 0.7), moderately suitable areas (0.5–0.7), low suitability areas (0.323064–0.5), and unsuitable areas (below 0.323064). To enhance the accuracy of distribution area statistics, we converted all model-calculated suitable distribution areas into the AlbersKrasovsky_1940_Albers projection, also referred to as the “Equal-Area Conic Projection” or “Double-Axis Latitude-Equalizing Conic Projection”. The jackknife technique was applied to evaluate the significance of each variable, and the predictive accuracy of our model was assessed through the area under the receiver operating characteristic curve (AUC). The closer the AUC value is to 1, the greater the model’s predictive precision.

## Figures and Tables

**Figure 1 plants-14-00541-f001:**
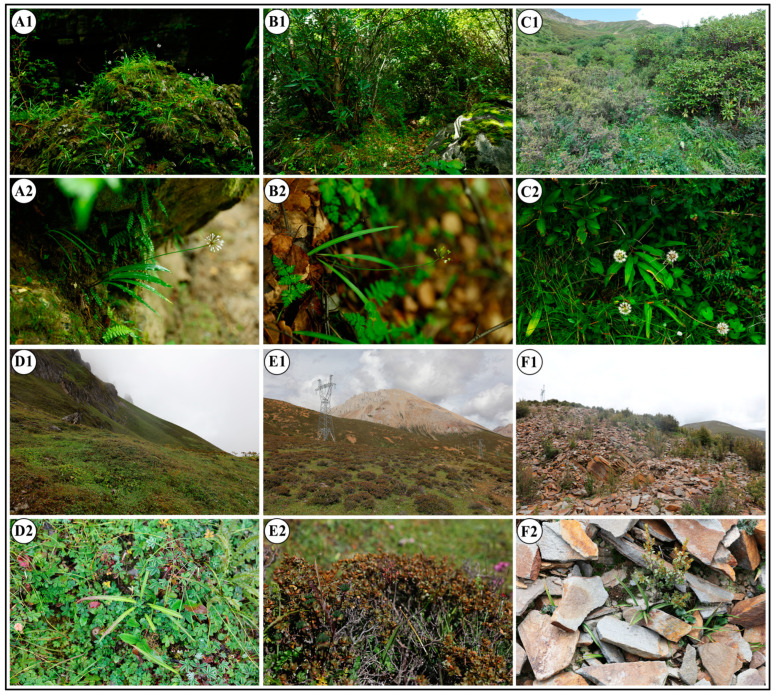
The diversified habitats and leaf characteristics of *A. prattii*. (**A1**,**A2**): Wangcang-A05; (**B1**,**B2**): Gongbujiangda-A18; (**C1**,**C2**): Maerkang-A06; (**D1**,**D2**): Gongshan-A35; (**E1**,**E2**): Deqing-A34; and (**F1**,**F2**): Bowa mountain-A42. For detailed population information, see Appendix A.

**Figure 2 plants-14-00541-f002:**
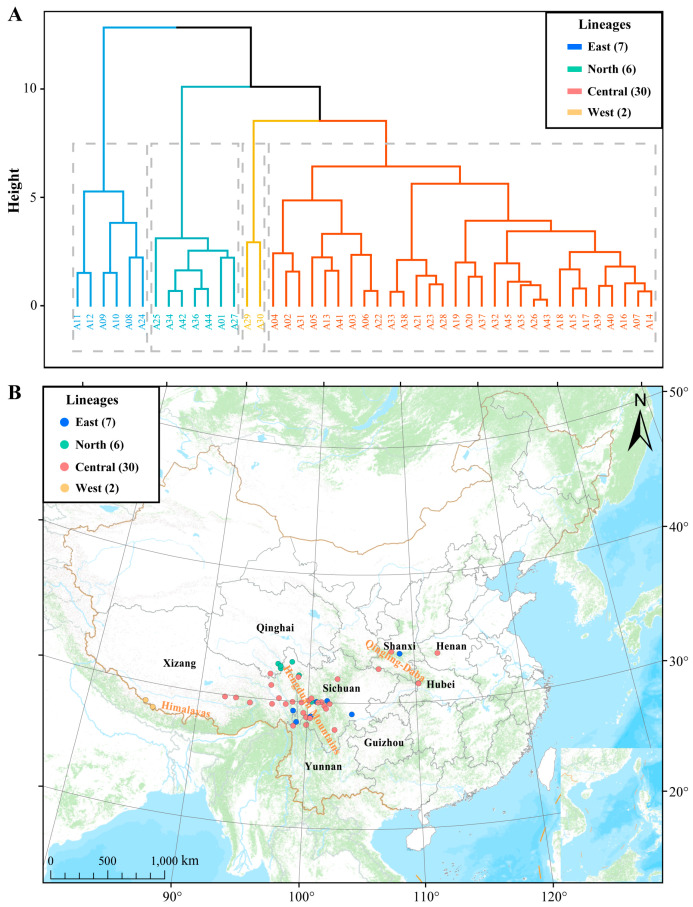
Hierarchical clustering of leaf morphological characteristics data in *A. prattii* populations. (**A**) The hierarchical clustering results of the morphological characteristics. (**B**) Geographical distribution of clustering results.

**Figure 3 plants-14-00541-f003:**
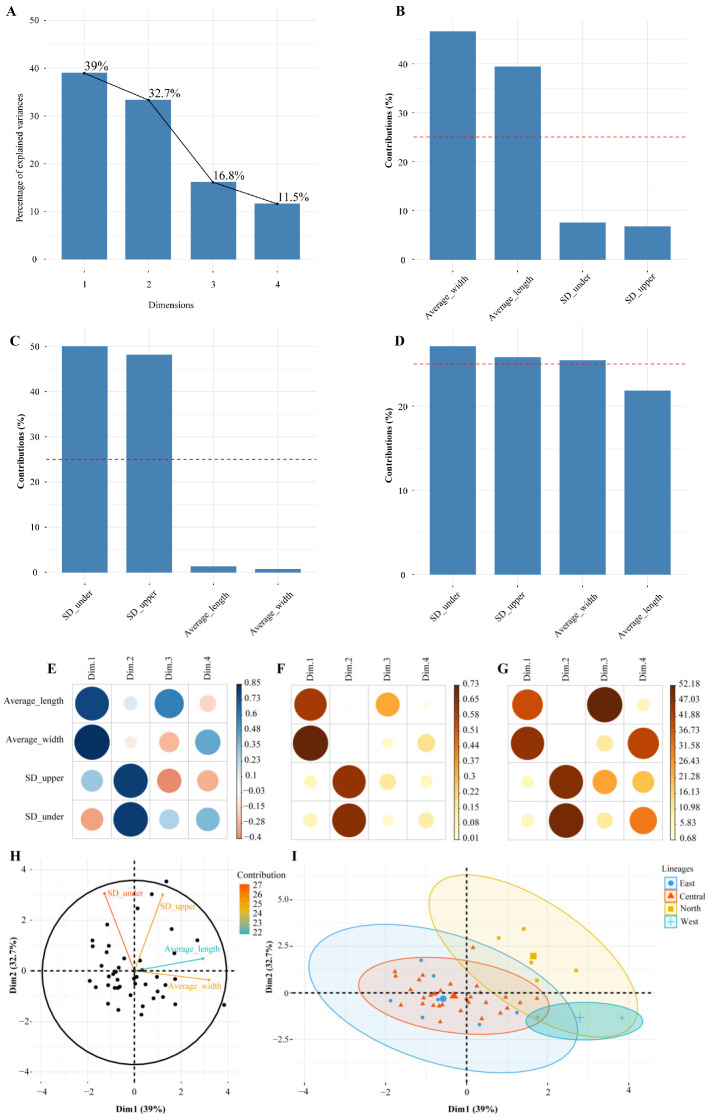
The principal component analysis is based on four leaf morphological characteristics. (**A**) Scree plot; (**B**) contribution of leaf characteristics to the first principal component; (**C**) contribution of leaf characteristics to the second principal component; (**D**) contribution of leaf characteristics to the principal components; (**E**) correlation between principal components and variables; (**F**) the squared correlation values between principal components and variables, with each row summing to 1; (**G**) the contribution of variables to the principal components; (**H**) the PCA results; and (**I**) the samples’ PCA results based on leaf characteristics.

**Figure 4 plants-14-00541-f004:**
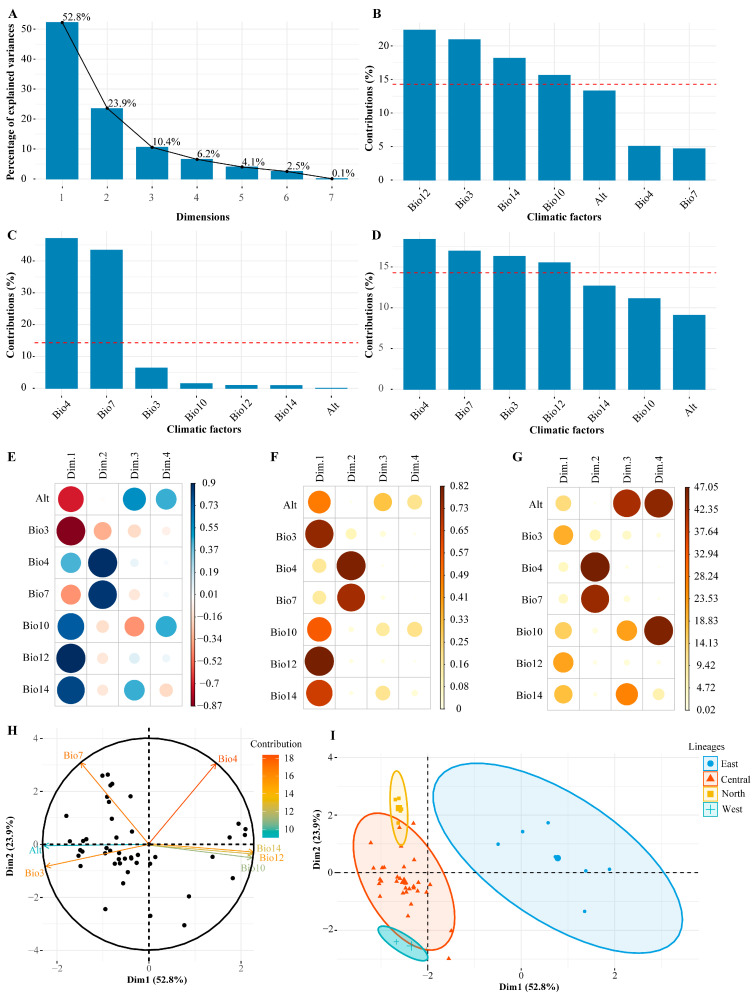
The principal component analysis is based on 7 major environmental factors. (**A**) Scree plot; (**B**) contribution of environmental variables to the first principal component; (**C**) contribution of environmental variables to the second principal component; (**D**) contribution of environmental variables to the principal components; (**E**) correlation between principal components and variables; (**F**) the squared correlation values between principal components and variables, with each row summing to 1; (**G**) the contribution of variables to the principal components; (**H**) the PCA results; and (**I**) the samples’ PCA results based on environmental factors.

**Figure 5 plants-14-00541-f005:**
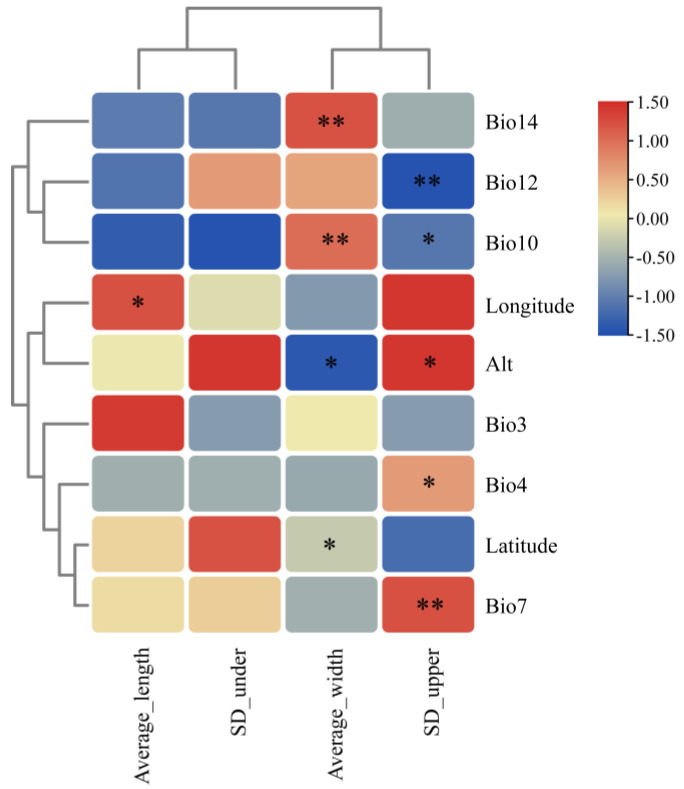
Heatmap of correlations between *A. prattii* morphological traits and major environmental factors (*: *p* < 0.05; **: *p* < 0.01).

**Figure 6 plants-14-00541-f006:**
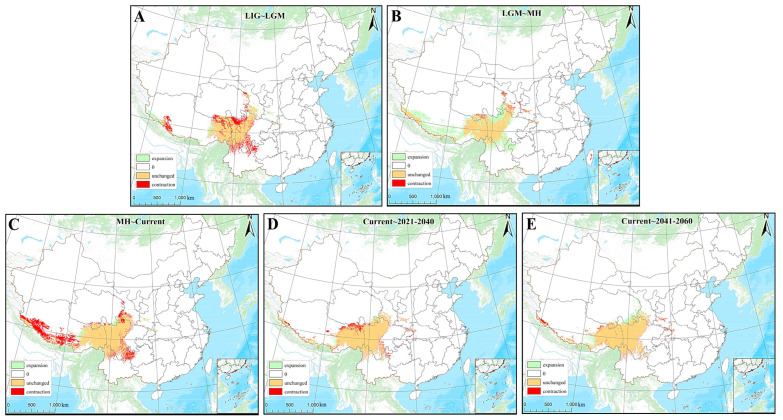
Distribution changes in *A. prattii* across different periods. (**A**) The distribution area changes in *A. prattii* from the LIG to the LGM; (**B**) the distribution area changes in *A. prattii* from the LGM to the MH; (**C**) the distribution area changes in *A. prattii* from the MH to the current period; (**D**) the distribution area of *A. prattii* from the current period to the future (2021–2040); and (**E**) the projected distribution area changes in *A. prattii* from the current period to the future (2041–2060).

**Figure 7 plants-14-00541-f007:**
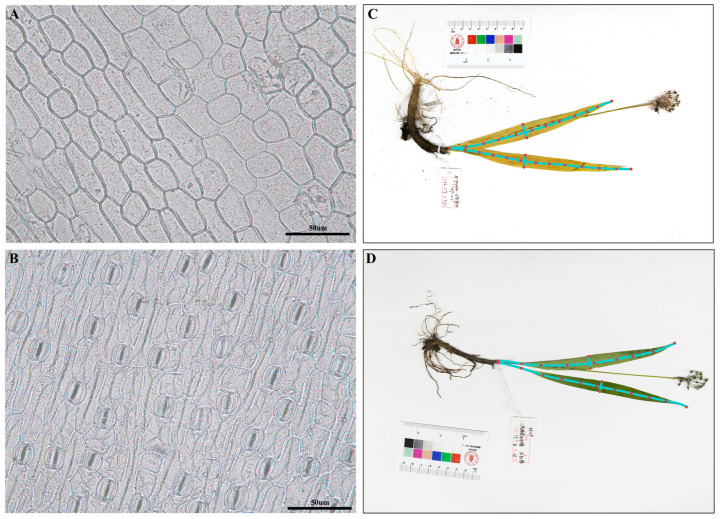
Leaf morphological characteristic observation and measurement. (**A**) The characteristics of the upper epidermis; (**B**) the characteristics of the under epidermis; and (**C**,**D**) examples of specimen morphological measurement.

**Table 1 plants-14-00541-t001:** Data statistics of leaf morphological and micro-morphological characteristics.

Population ID	Average Leaf Length	VC (%)	Average Leaf Width	VC (%)	Stomatal Density Upper Epidermis (/mm^2^)	Stomatal Density Under Epidermis (/mm^2^)
A01	25.86	13.64	1.11	20.41	0	81.25
A02	23.59	14.58	0.94	33.26	0	45.83
A03	21.93	33.03	3.12	24.82	0	34.38
A04	12.55	13.66	1.49	27.69	0	27.08
A05	23.53	13.22	1.52	21.3	0	9.38
A06	24.4	20.5	2.3	21.72	0	29.17
A07	24.99	18.5	1.57	19.4	0	42.71
A08	35.08	17.1	2.07	31.28	12.5	68.75
A09	36.19	5.87	2.45	34.7	19.79	39.58
A10	27.79	14.44	2.34	39.49	17.71	34.38
A11	21.8	9.72	2.02	21.65	31.25	68.75
A12	24.97	12.55	2.63	21.92	30.21	85.42
A13	22.92	6.27	1.81	26.79	0	19.79
A14	24.43	27.88	1.67	13.13	0	48.96
A15	19.85	4.14	1.42	26.7	0	41.67
A16	29.21	21.7	1.52	31.31	0	39.58
A17	16.85	19.84	1.7	46.61	0	42.71
A18	17.5	7.2	0.53	18.19	0	39.58
A19	20.6	6.55	1.01	37.53	13.54	38.54
A20	21.87	19.6	0.81	42.63	8.33	39.58
A21	31.43	29.94	0.91	32.67	0	44.79
A22	26.85	12.29	2.3	16.65	0	35.42
A23	32.48	19.57	1.79	18.45	0	46.88
A24	29.9	18.01	0.87	30.13	19.79	72.92
A25	17.64	4.96	1.24	29.57	12.5	80.21
A26	22.09	8	1.16	22.03	0	56.25
A27	25.1	10.15	2.2	25.6	0	69.79
A28	28.37	7.58	1.71	3.23	0	56.25
A29	32.28	2.4	4.66	27.81	0	25
A30	30.74	10.86	2.44	40.78	0	19.79
A31	16.5	47.71	0.47	37.58	0	46.88
A32	14.98	8.78	1.7	24.12	0	56.25
A33	35.03	9.65	1.73	20.84	0	36.46
A34	19.77	12.95	0.9	17.85	0	90.63
A35	20.24	11.43	0.83	28.98	0	62.5
A36	23.29	10.31	0.93	15.91	0	76.04
A37	15.86	25.27	1.08	49.42	4.17	34.38
A38	36.22	18.91	1.92	19.45	3.13	35.42
A39	22.35	5.63	1.48	24.31	0	31.25
A40	22.02	4.06	0.86	30.18	0	37.5
A41	14.41	28.3	1.65	24.06	0	19.79
A42	17.08	39.7	1.16	20.9	0	86.46
A43	21.84	8.76	1.23	18.42	0	56.25
A44	24.87	8.48	0.94	23.06	0	82.29
A45	24.36	6.45	0.73	25.06	0	51.04

VC: coefficients of variation (average values/standard deviation × 100%).

## Data Availability

All data involved in this study have been included in the text and tables.

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
