# Peer review of "Investigating the Variation in Leaf Traits Within the Allium prattii C.H. Wright Population and Its Environmental Adaptations"

_plants, 2025, doi:10.3390/plants14040541_

Round 1

Reviewer 1 Report

Comments and Suggestions for Authors

I have read the paper and overall, I found it to be well-written and informative.

The introduction is comprehensive and well-structured, although it could benefit from including more examples.

The materials and methods section is detailed and clearly written.

The results are clearly presented. However, the authors could perform some tests to determine if there are differences between populations and lineages. Additionally, the authors have tested correlations between environmental and morphological variables, but they have not tested correlations between longitude/latitude and morphological variables. In other words, are geographically closer populations morphologically more similar, while geographically distant populations are more different? Furthermore, Table 2 can be included as supplementary material.

Additionally, it would be useful to quantify the variability of populations in some way, for example, by calculating coefficients of variation.

In the discussion, it would be beneficial to elaborate more on the variability among and within populations. For instance, are some populations more variable than others? Are there differences in variability between populations from different altitudes?

Additional minor corrections and comments are provided in the attached document.

Comments on the Quality of English Language

The English language used in the paper is generally good and does not hinder understanding of the research. However, there are instances where the choice of words could be improved to enhance clarity and precision. Minor revisions are needed to ensure the use of the best possible expressions throughout the paper.

Author Response

Dear reviewer,

Thank you very much for your comments and professional advice concerning our manuscript entitled “Investigating the variation in leaf traits within the Allium prattii population and its environmental adaptations” (Manuscript ID: plants-3394374). Your insights greatly contribute to enhancing the academic rigor of our manuscript. Building upon your suggestions, we have conducted a thorough revision, which includes making additional analyses, adjusting the Tables in text, and revising the English in the whole text . In particular, we have meticulously revised sections that involved in the morphological variability among populations and the correlations between longitude/latitude and morphological variables. Below, we provide detailed changes, which are also tracked with red words in the revised manuscript. We sincerely appreciate your diligent efforts and hope that the revisions meet with approval. Our responses to your questions are as follows:

  1. The introduction is comprehensive and well-structured, although it could benefit from including more examples.

Response: Thanks very much for your comments, your suggestions have been immensely helpful in revising and improving our manuscript. Following your advice, we have added more examples and corresponding references in the introduction, and we hope the revised version can meet your approval.

  1. The materials and methods section is detailed and clearly written.

Response: Thanks for your comments, we also have checked each section of our manuscript.

  1. The results are clearly presented. However, the authors could perform some tests to determine if there are differences between populations and lineages. Additionally, the authors have tested correlations between environmental and morphological variables, but they have not tested correlations between longitude/latitude and morphological variables. In other words, are geographically closer populations morphologically more similar, while geographically distant populations are more different? Furthermore, Table 2 can be included as supplementary material.

Response: We sincerely appreciate your valuable comments, yes, it is important to detect the correlations between geographical distribution and morphological variables. In our revised manuscript, we have further integrated the analysis of latitude and longitude, along with an examination of the morphological traits of species populations. Additionally, we have explored the correlations between a range of climatic variables, altitude, geographical distribution, and the morphological variations within populations. Additionally, we have adjusted the Table in the text, and removed the Table 2 into supplementary materials. We believe that our manuscript has been greatly improved and upgraded, and we hope that it will meet your approval.

  1. Additionally, it would be useful to quantify the variability of populations in some way, for example, by calculating coefficients of variation.

Response: Thanks for your suggestion, in accordance with your recommendation, we initially calculated the mean values of leaf morphological indices for each plant within each population. Subsequently, we determined the standard deviations of these leaf morphological data indices for each population. Ultimately, we assessed the coefficient of variation for each leaf morphological indicator within each population by computing the percentage of the standard deviation relative to the mean value. The detailed information have been added in the revised Table 1. hope our revision can get your approval.

  1. In the discussion, it would be beneficial to elaborate more on the variability among and within populations. For instance, are some populations more variable than others? Are there differences in variability between populations from different altitudes?

Response: Yes, the comparative analysis of morphological traits among populations is crucial. In our revised discussion, we examined the interplay between population morphological characteristics and factors such as geography, altitude, and climate. We investigated the potential correlations between morphological variations in plant populations and their geographical distribution and altitude. Furthermore, we explored in depth the adaptive evolutionary patterns of A. prattii as influenced by diverse climatic conditions across different geographical areas.

  1. Additional minor corrections and comments are provided in the attached document.

Response: We are immensely grateful for your meticulous comments. We have thoroughly revised our work in accordance with your insightful comments and are hopeful that our revised version will meet with your approval.

  1. The English language used in the paper is generally good and does not hinder understanding of the research. However, there are instances where the choice of words could be improved to enhance clarity and precision. Minor revisions are needed to ensure the use of the best possible expressions throughout the paper.

Response: OK, we have made a deep revision of the English text according to your suggestions. We hope our new version can meet your approval.

Thanks once again for your comments, please do not hesitate to contact me with any additional questions. We look forward to your correspondence.

Sincerely

Deng-Feng Xie

E-mail address: df_xie2017@163.com.

Reviewer 2 Report

Comments and Suggestions for Authors

The manuscript submitted for review regarding the morphology of Allium pratii can be published in Plants. The text is written very well and clarifies the issues the authors address. This is a very detailed and well-prepared study. I have only very small comments because of the numbering of the tables and one quote. See my comments in pdf text.

Author Response

Dear reviewer,

Thank you very much for your comments and professional advice concerning our manuscript entitled “Investigating the variation in leaf traits within the Allium prattii population and its environmental adaptations” (Manuscript ID: plants-3394374). Your insights greatly contribute to enhancing the academic rigor of our manuscript. Building upon your suggestions, we have conducted a thorough revision, which includes making additional analyses, adjusting the Tables in text, and revising the English in the whole text . In particular, we have meticulously revised sections that involved in the morphological variability among populations and the correlations between longitude/latitude and morphological variables. Below, we provide detailed changes, which are also tracked with red words in the revised manuscript. We sincerely appreciate your diligent efforts and hope that the revisions meet with approval. Our responses to your questions are as follows:

Main comments:

The manuscript submitted for review regarding the morphology of Allium pratii can be published in Plants. The text is written very well and clarifies the issues the authors address. This is a very detailed and well-prepared study. I have only very small comments because of the numbering of the tables and one quote. See my comments in pdf text.

Response: We are deeply grateful for your invaluable comments. Following your suggestions, we have reorganized the sequence of the tables and thoroughly revised the entire text. We sincerely hope that our revised version meets with your approval.

Thanks once again for your comments, please do not hesitate to contact me with any additional questions. We look forward to your correspondence.

Sincerely

Deng-Feng Xie

E-mail address: df_xie2017@163.com.

Round 2

Reviewer 1 Report

Comments and Suggestions for Authors

Review of the article: "Investigating the variation in leaf traits within the Allium prattii C.H. Wright population and its environmental adaptations"

I have read the revised version of the article. The authors have addressed some of the issues raised in the previous review, but there are still several points that need to be improved.

Differences Between Groups: The authors mention in the discussion that certain groups differ significantly. However, there are no tests mentioned in the materials and methods or results sections to support these claims. It is essential to test for homogeneity of variances and differences between groups. If variances are homogeneous, use ANOVA; if not, use the Kruskal-Wallis test.

Correlation Analysis: The correlation analysis includes geographical variables such as latitude and longitude, which were significantly correlated with certain leaf traits. This should be discussed in more detail. Additionally, these variables (longitude and latitude) should be mentioned in the abstract and in the materials and methods sections.

Discussion Structure: Some parts of the discussion boil down to the repetition of the results, such as the second paragraph in the discussion section. The discussion should be structured according to the research objectives. Either expand the research objectives in the introduction or reduce the number of subheadings in the results and discussion sections to improve the structure and clarity of the paper.

Coefficient of Variation: Please use CV for the coefficient of variation. In Table 1, CV values should be placed in separate columns.

Overall, the article has improved, but further revisions are necessary to ensure clarity and scientific rigor.

Author Response

Dear reviewers,

We sincerely appreciate the constructive and insightful comments you have provided, which have been of great guidance to us. Following your suggestions, we have thoroughly revised the manuscript once again. Specifically, we have conducted an ANOVA test to examine the morphological differences among groups, supplemented the unclear descriptions in the Methods and Results sections, and revised some redundant and structurally ambiguous parts. We hope that our revised version will meet with your approval. Our responses to your questions are as follows:

  1. I have read the revised version of the article. The authors have addressed some of the issues raised in the previous review, but there are still several points that need to be improved.Differences Between Groups: The authors mention in the discussion that certain groups differ significantly. However, there are no tests mentioned in the materials and methods or results sections to support these claims. It is essential to test for homogeneity of variances and differences between groups. If variances are homogeneous, use ANOVA; if not, use the Kruskal-Wallis test.

Response: We sincerely apologize for not noticing this very important point, which is very important for explaining the morphological differences among different groups. Following your advice, we performed the ANOVA test and examined the morphological differences among groups, the test results can be achieved in Figure S1. We also added related information in methods and results, hoping our revised version can meet your approval.

  1. Correlation Analysis: The correlation analysis includes geographical variables such as latitude and longitude, which were significantly correlated with certain leaf traits. This should be discussed in more detail. Additionally, these variables (longitude and latitude) should be mentioned in the abstract and in the materials and methods sections.

Response: Thanks for your suggestions, in the revised version, we added the information involved in the correlation between the seven climatic factors, longitude, latitude and leaf morphological characters in the abstract, materials and methods. Meanwhile, we also made detailed discussion on this correlation.

  1. Discussion Structure: Some parts of the discussion boil down to the repetition of the results, such as the second paragraph in the discussion section. The discussion should be structured according to the research objectives. Either expand the research objectives in the introduction or reduce the number of subheadings in the results and discussion sections to improve the structure and clarity of the paper.

Response: Thank you very much for your suggestions, which have been extremely helpful in improving our manuscript. Following your advice, we have made adjustments to the structure and content of our manuscript, ensuring that the overall structure and logic are clear and that the content is appropriately arranged. We hope that our revised version will meet with your approval.

  1. Coefficient of Variation: Please use CV for the coefficient of variation. In Table 1, CV values should be placed in separate columns.Overall, the article has improved, but further revisions are necessary to ensure clarity and scientific rigor.

Response: Ok, we have made revision for this point, we also checked and revised all the Tables and Figures in our manuscript.

Thanks once again for your comments, please do not hesitate to contact me with any additional questions. We look forward to your correspondence.

Sincerely

Deng-Feng Xie

E-mail address: df_xie2017@163.com.

Round 3

Reviewer 1 Report

Comments and Suggestions for Authors

The article has been significantly improved and can be accepted for publication.